# Validation and Demonstration of an Atmosphere-Temperature-pH-Controlled Stirred Batch Reactor System for Determination of (Nano)Material Solubility and Dissolution Kinetics in Physiological Simulant Lung Fluids

**DOI:** 10.3390/nano12030517

**Published:** 2022-02-02

**Authors:** Else Holmfred, Katrin Loeschner, Jens J. Sloth, Keld Alstrup Jensen

**Affiliations:** 1National Research Centre for the Working Environment, 2100 Copenhagen, Denmark; 2Research Group for Analytical Food Chemistry, Division of Food Technology, National Food Institute, Technical University of Denmark, 2800 Kgs. Lyngby, Denmark; kals@food.dtu.dk (K.L.); jjsl@food.dtu.dk (J.J.S.)

**Keywords:** Nanomaterials, abiotic in vitro testing, physiological fluids, batch reactor, inductively coupled plasma-mass spectrometry

## Abstract

In this study, we present a dissolution test system that allows for the testing of dissolution of nano- and micrometer size materials under highly controlled atmospheric composition (O_2_ and CO_2_), temperature, and pH. The system enables dissolution testing in physiological simulant fluids (here low-calcium Gamble’s solution and phagolysosomal simulant fluid) and derivation of the temporal dissolution rates and reactivity of test materials. The system was validated considering the initial dissolution rates and dissolution profiles using eight different materials (γ-Al_2_O_3_, TiO_2_ (NM-104 coated with Al_2_O_3_ and glycerin), ZnO (NM-110 and NM-113, uncoated; and NM-111 coated with triethoxycaprylsilane), SiO_2_ (NM-200—synthetic amorphous silica), CeO_2_ (NM-212), and bentonite (NM-600) showing high intra-laboratory repeatability and robustness across repeated testing (I, II, and III) in triplicate (replicate 1, 2, and 3) in low-calcium Gamble’s solution. A two-way repeated-measures ANOVA was used to determine the intra-laboratory repeatability in low-calcium Gamble’s solution, where Al_2_O_3_ (*p* = 0.5277), ZnO (NM-110, *p* = 0.6578), ZnO (NM-111, *p* = 0.0627), and ZnO (NM-113, *p* = 0.4210) showed statistical identical repeatability across repeated testing (I, II, and III). The dissolution of the materials was also tested in phagolysosomal simulant fluid to demonstrate the applicability of the ATempH SBR system in other physiological fluids. We further show the uncertainty levels at which dissolution can be determined using the ATempH SBR system.

## 1. Introduction

Manufactured nanomaterials (NMs) have increasingly been produced for a few decades [1,2] and are used in multiple industrial sectors [3,4,5,6] as nanotechnology inspires new solutions and products [2,5,7]. Compared to the bulk material, NMs demonstrate distinct properties utilized to solve existing problems (e.g., material durability and strength, rheology, catalysis, optics, drug delivery, and food packing) [3,4,5,6]. In Europe, the European Commission adopted a recommendation on the definition of nanomaterial in 2011 where a NM generically is defined as a material in which at least 50% of the particles in number size distribution (in the unbound state or as an aggregate or as an agglomerate) have one or more external dimensions in the size range of 1–100 nm [8]. 

In the recommendation, a “particle” is defined as a minute piece of matter with defined physical boundaries, which could, e.g., be spheres, flakes, and fibers. This definition is applied for defining substances in nanoform in the European chemical regulations [9]. NMs may have higher human [2] and environmental toxicity [10,11] when compared to larger-size materials of the same compounds. The use of nanotechnology is expected to increase in the coming years [12,13], although the tools for predicting potential toxicities are still limited and under discussion.

Physicochemical parameters, such as the solubility and dissolution rate, are critical parameters in different industries, including the pharmaceutical [14,15], food [16], and cosmetic fields [17], and plays an important role in risk assessment as well as grouping and read-across [18,19]. Dissolution is a dynamic process fundamentally controlled by the thermodynamic parameter of solubility, which, along with the concentration gradient, acts as the driving force for the dissolution of the material [20]. 

Considering the properties of only the NM, the size and thereby the surface area are the primary physicochemical parameters affecting the solubility, dissolution, and dissolution rate. However, the crystallinity, morphology, and surface-chemical modifications also influence the solubility of NMs [20,21]. Agencies, such as The European Chemicals Agency (ECHA) as well as the Organisation for Economic Co-operation and Development (OECD), provide guidelines that focus on the characterization and testing of chemicals including nanoforms [22]. Both the OECD and ECHA identify information on solubility and dissolution rates in relevant media as fundamental for the read across and assessment of the potential risks of NMs [9,22,23].

The potential bioavailability of constituent elements and residence time of materials can be estimated by studying the dissolution behavior in a physiological fluid. As seen in the European Pharmacopoeia (Ph. Eur.) and the United States Pharmacopoeia (USP), dissolution testing is often a legal requirement for drug approval using the harmonized basket, paddle, reciprocating cylinder, or flow-through system [24,25]. 

The typical Ph. Eur. and USP dissolution test medium is water; however, buffers in the physiological range (pH 1.2–7.5) and diluted acids are also recommended to mimic the gastrointestinal biodissolution of drugs [24,25]. In the 1980s and 1990s, the potential health effects of inhaled asbestos and man-made fibers were heavily studied [26]. The in-vitro dissolution in simulated lung fluids was used to evaluate the biosolubility, as the residence time in the lung was evaluated as one of the important physicochemical effects influencing the risk of disease [27,28,29].

There are no international standard procedures for testing the dissolution of NMs in physiologically relevant fluids, which is an immediate requirement due to recent changes in regulatory information requirements on NMs regarding both chemicals and food [30]. Dissolution kinetics (and solubility) are requested to group and read-across different nanoforms and support toxicological findings in biodurability [31,32,33] and eventually can reduce animal testing needs. 

Consequentially, dissolution studies must be conducted in physiologically relevant simulant fluids, as the predictability and comparability with in vivo systems otherwise would risk being inadequate. As an example, nanosized ZnO demonstrates low solubility in water [34], and Avramescu et al. (2016) showed how the solubility of ZnO was affected by changes in pH. At low pH, the solubility of both nanosized ZnO (<50 nm) and the bulk analogue was significantly higher (approximately 48 fold) than at neutral pH conditions [35].

Currently, the ability to predict the hazards of NMs through the correlation between toxicological endpoints and physicochemical properties of NMs, including the solubility and dissolution, is restricted [32]. Dissolution studies (and the obtained rate constants) can potentially be essential experiments for read-across, grouping, and assessment of biokinetic behavior and potential hazards [31,33,36].

Exposure to NMs through pulmonary airways is generally considered the exposure route of highest concern. Repeated incidental exposure of NMs is observed for workers in both the development and production and use of NMs [37,38,39,40]. A substantial fraction of inhaled NMs deposit in the deep and sensitive bronchoalveolar region of the lung, where an accumulation of particles can occur and cause severe toxic effects [36,41]. The observed toxicity is, among other factors, dependent on the solubility, dissolution rate, and the reactivity of the deposited material. 

Poorly soluble NMs have been found to accumulate in lung tissue, causing potential long-term effects [42], while more rapidly dissolving NM may or may not cause acute toxic effects depending on chemical composition [41]. However, manufactured materials increasingly become more advanced and are often no longer simple mono-substance materials. Materials that are coated or doped with organic and/or inorganic compounds add complexity to this puzzle, and it is difficult to predict the dissolution behavior of such complex materials using basic in silico modelling approaches.

To mimic NM behavior after pulmonary exposure, simulated lung fluids, such as Gamble’s solution (simulant for the lung lining fluid) [27] and the phagolysosomal simulant fluid (simulant for the alveolar macrophage fluid) [43] were found to be adequate for simple acellular in vitro testing [44]. Koltermann-Jülly et al. (2018) [31] and Keller et al. (2020) [45] have both studied the dissolution behavior of NMs in phagolysosomal fluid (PSF) using a continuous flow-through system. The abiotic dissolution system showed the ability to differentiate between fast, partial, and slow dissolving materials and to determine the dissolution rates with good correlation to in vivo studies in the case of BaSO_4_ [45]. 

Batch reactors were previously used to study dissolution kinetics of mineral fibers [46], and a similar static pH-controlled batch reactor was used to study the digestion of pharmaceuticals [47]. To the best of our knowledge, an atmosphere-temperature-pH-controlled stirred batch reactor system has not previously been used to study the dissolution kinetics of NMs. The advantages of this dissolution system include not only the tight control of pH, temperature, and gas flow (O_2_ and CO_2_) but also the possibility to study the real-time effect on the redox potential in the media caused by the test materials.

The reactivity of a material in relevant media is another important measure to consider and includes oxidative stress in exposed tissue. The release of free electrons (oxidation of NMs) can result in formation of reactive oxygen species associated with cell and DNA damage [48]. Redox potential (*E_h_*) is one possible measure for describing the oxidative reactivity of a NM. 

Conducting dissolution testing under comparable conditions can help to understand to which degree a deposited NM influences the natural *E_h_*-pH range and therefore support the understanding of the potential reactivity of NMs. Plumlee and Ziegler (2003) described the different biological compartments and their specific natural *E_h_*-pH range [48]. In general, human fluids will, as an effect of composition and concentration, naturally vary in the redox potential. PSF acts as a simulant of macrophage lysosome, and lysosomes have been described with natural variation from −50 to 160 mV [48].

The aim of this work was to test and document an atmosphere-temperature-pH-controlled stirred batch reactor system (ATempH SBR) with online redox potential measurement for short-term *abiotic* in vitro dissolution testing of NMs in physiological relevant fluids (low-calcium Gamble’s solution and phagolysosomal fluid; PSF). The ATempH SBR system was intra-laboratory validated using eight different materials with different expected dissolution rates (fast, partial, and slow dissolution), and the materials were well-characterized benchmark materials originating from a.o. the OECD working party on NMs sponsorship program [49].

## 2. Materials and Methods

### 2.1. Nanomaterials

TiO_2_ (NM-104), ZnO (NM-110, NM-111, and NM-113), SiO_2_ (NM-200), CeO_2_ (NM-212), and bentonite (NM-600) were all obtained from the Fraunhofer Institute for Molecular Biology and Applied Ecology (Schmallenberg, Germany). Gamma (γ-Al_2_O_3_ was purchased from IoLiTec Ionic Liquids Technologies GmbH (Heilbronn, Germany) and the subsamples were supplied by from Bundesinstitut für Risikobewertung, BfR (Berlin, Germany). 

NM-104 is a rutile coated with Al_2_O_3_ and glycerin. The ZnO materials are all zincite, of which NM-111 is coated with triethoxycaprylsilane. NM-200 is synthetic amorphous silica, while NM-212 is cerianite. NM-600 is a natural clay material mainly consisting of montmorillonite. The NMs, except γ-Al_2_O_3_, were stored under argon before use and in a desiccator after subsampling to prevent sorption of the humidity from the air.

### 2.2. Wavelength Dispersive X-ray Fluorescence Spectroscopy

Powder materials were pelletized using Cereox matrix (20 wt.% for all except 60% for SiO_2_ (NM-200)) and analyzed (Be-U) by Wavelength Dispersive X-ray Fluorescence (WDXRF) using a Bruker S8 Tiger using PET, LIF 200, XS55 analyzer crystals (Billerica, MA, USA). The analysis was performed using the Quant Express method quantifying the elemental concentrations using instrument standards. The obtained spectra were carefully analyzed for potential peak overlaps before final quantification of the samples was made as un-normalized oxides. Detection limits for the different oxides typically varied from ca. 15 µg/g to ca. 500 µg/g for trace and minor elements.

### 2.3. Thermogravimetric Analysis

The materials were analyzed one to three times by coupled Thermogravimetric analysis (TGA) Mass Spectrometry (MS) using a Netzsch STA 449 F3 Jupiter and a QMS D Aëolos mass spectrometer (NETZSCH-Gerätebau GmbH, Selb, Germany), respectively. The TGA was run using 40% air and 60% of nitrogen by volume and the temperature program adopted from previous work described in Clausen et al. (2019) [50]: Heating from room temperature to 50 °C at 10 °C/min and holding for 1 min, then heating to 100 °C at 2.5 °C/min and hold for 10 min, then heating to 800 °C at 2.5 °C/min and hold for 1 min, followed by cooling down to room temperature. 

The crucibles had a volume of 3.4 mL and were made of alumina (Al_2_O_3_). Samples were taken from the sample vials and analyzed directly after weighing, and further conditioning was not made to equilibrate with known air humidity. Data were corrected for buoyancy. Mass losses that occurred between room temperature and 100 °C were ascribed to moisture content while mass losses at higher temperatures were ascribed to organic coatings, hydroxyl groups, or other associated degradable materials and given as loss-on-ignition (LOI). The results were used to calculate the true amount of test material dissolved.

### 2.4. X-ray Diffraction

X-ray diffraction (XRD) analysis was completed on the samples for analysis of crystalline phase(s) and potential impurities using Bruker D2 Phaser (30 kV; 10 mA; 1.548 Å Cu K-line) equipped with a LYNXEYE_EX_T detector (Billerica, MA, USA) (1D-mode). The optic parameters were 4° soller slits, 1.0 mm divergence slits, 8 mm antiscatter slit, and a 1 mm knife. Smear analysis was made on bentonite (NM-600) using 2.3° and 2.5° soller slits, 3.0 mm antiscatter slit, and a 1 mm knife. 

The bentonite (NM-600) smear was made after dispersion in ethanol and was made to gain better data on impurities. Scans were made in a continuous PSD (position sensitive detector) fast scan coupled °2Theta mode from 0 to 110 °2Theta and step size of 0.02 °2Theta with 0.75 s/step. The sample holder was set to rotate 10–30 rpm. All phase-identifications were performed using the EVA software. All samples were prepared in a front-loaded sample holder.

### 2.5. Physiological Relevant Fluids

Phagolysosomal simulant fluid (PSF) was prepared by dissolving the components of Table 1 in 2 L ultrapure water (18 MΩ·cm at 25 °C) (Thermo Fisher Scientific, Waltham, MA, USA). The solution was left overnight and filtered the following day through a polyvinylidene fluoride membrane 0.45 µm filter (Merck Millipore Ltd., Tullagreen, Ireland). PSF has a shelf-life of approximately 1–1.5 months stored at 5 °C protected from light. All chemicals were purchased from Merck (Darmstadt, Germany).

Low-calcium Gamble’s solution as a simulant for the lung-lining fluid was prepared by dissolving the components of Table 2 in 2 L ultrapure water. The solution was ultrasonicated for 30 min, left overnight, and the following day filtered through a polyvinylidene fluoride membrane 0.45 µm filter (Merck Millipore Ltd., Tullagreen, Ireland). Low-calcium Gamble’s solution has a shelf-life of approximately 1–1.5 months when stored at 5 °C and protected from light. All chemicals were purchased from Merck (Darmstadt, Germany).

### 2.6. Dispersion of Nanomaterials

The NMs were dispersed following the NANoGENOTOX batch dispersion protocol validated as part of the FP7 NANoREG project [51]. Before dispersion, a 0.05% *w*/*v* bovine serum albumin (BSA) solution was prepared in ultrapure water (18 MΩ·cm, 21 °C, Thermo Fisher Scientific, Waltham, MA, USA). BSA (obtained from Sigma–Aldrich (now Merck), Darmstadt, Germany) was dissolved in ultrapure water to obtain a 1% *w*/*v* solution, stored overnight, and sterile-filtered (0.22 µm). The 1% *w*/*v* BSA solution was diluted to 0.05% *w*/*v* with ultrapure water.

We weight 37.5 mg of NM into a 15 mL Scott-Durham glass vial, and pre-wetted with 75 µL 96% ethanol (Merck, Darmstadt, Germany), followed by dispersed with 14.57 mL 0.05% *w*/*v* BSA solution to a final concentration of 2.56 mg/mL. If the weighted NM differed from 37.5 mg, the volumes were adjusted to obtain exactly 2.56 mg/mL dispersion concentration. A 400 W Branson Sonifier S-450D (Branson Ultrasonics Corp., Danbury, CT, USA) equipped with a 13 mm disruptor horn was used to sonicate the particle dispersion directly after adding the suspension media for 16 min with a 10% amplitude (approximately 42 W). The sonication was performed under constant cooling in an ice-water bath.

### 2.7. Dynamic Light Scattering and Laser Doppler Electrophoresis

The mean hydrodynamic size (z-average, Z_ave_), estimated width of the size distribution (polydispersity index, PDI), size distribution, and zeta potential (ζ_pot_) were determined to evaluate the quality and state of particle dispersion. The results were obtained using a Malvern Zetasizer Nano ZS (Malvern Panalytics Ltd., Malvern, United Kingdom) device equipped with a 633 nm laser using 173° as the measurement angle for non-invasive backscattering measurements. 

Immediately after dispersion with the sonicator, 700 µL of the particle dispersion was transferred to a disposable folded capillary cell (DTS1070, Malvern Panalytics) and analyzed after 5 min thermal equilibration. Measurements were conducted at 25 °C using the viscosity of water (0.8872 cP) with an equilibration time of 120 s. The size distribution was measured using polystyrene latex as reference with optical index 1.590. The Z_ave_, PDI, size distribution and zeta potential were reported as an average of ten repeated measurements. The zeta potential was calculated in automatic mode using the Smoluchowski model [52] based on ten repeated sample measurements (*n* = 1). The percentage number distributions are found in Appendix A.

### 2.8. Atmosphere-Temperature-pH-Controlled Stirred Batch Reactor System

The dissolution studies were performed using the ATempH SBR system. The ATempH SBR system consists of four separate identical reactor units. One unit is used as a reference containing the pure test medium (in this case, PSF or low-calcium Gamble’s solution). In contrast, the other three units (replicates 1–3) are used for replicate dissolution testing of the NMs (*n* = 3). Each reactor unit has a separate OMNIS titration module (Metrohm, Herisau, Switzerland) with two liquid adaptors for continuous pH adjustment with 1 M HCl and 1 M NaOH (Reagecon Diagnostics Ltd., Country Clare, Ireland). 

The titration volume (recorded every 10 s) was used to calculate the acid/base dilution between each sampling time point (adding between 0.5–1.2 mL acid/base during the 24 h dissolution study, depending on the NM and test medium). The 120 mL double-walled glass reactors allowed for a constant temperature of 37 °C by using a PolyScience water pump (Holm & Halby, Brøndby, Denmark) to circulate heated water continuously. To protect the NMs from light, each reactor was gently wrapped in aluminum foil. Features of the SBR system are illustrated in Figure 1.

Each reactor is equipped with a pH-electrode to regulate the titration modules and a Pt redox electrode for potential data collection of NM reactivity (Metrohm, Herisau, Switzerland). Before testing, the Pt redox electrodes were calibrated with a 250 mV solution, and the pH electrodes were calibrated with pH 4.0 and 7.0 solutions (Reagecon Diagnostics Ltd., Clare, Ireland). When mimicking lung conditions, the SBR system was mounted with a MultiFlo Cable Kit mass-flow meter and controller (Brooks Instrument, Hatifield, PA, USA) regulating the gas flow of CO_2_ and O_2_ at 5.62 and 144 mL/min, respectively. The adjustable speed of the three-bladed propeller stirrer was set to 840 rpm throughout all experiments. 

The selected NMs were tested in both PSF and low-calcium Gamble’s solution with a test volume of 96 mL simulated lung fluid in each reactor. To test the repeatability of the SBR system, all particles tested in Gamble’s solution were tested on three different days (three repeats × three replicates = nine dissolution tests) denoted I, II, and II. Additional testing was performed in PSF using one repeat for comparison of the different test media (one repeat × three replicates = three dissolution tests). Prior to testing, the low-calcium Gamble’s solution and PSF are adjusted to pH 7.4 and 4.5, respectively, and kept constant throughout testing.

The particle dispersion was transported to testing immediately after completion of the probe sonication. To ensure the best possible dispersion for dosing, the suspensions were vortexed approximately 10 s before 4 mL of the dispersion was added to each of the three of the reactors filled with 96 mL simulant fluid using a pipette, creating a nominal starting concentration of 102.4 mg/L in 100 mL. The blank reactor was added 4 mL of the batch dispersion medium to 96 mL simulant fluid. 

At selected time points; t_sampling_ = t_0_, t_1_, t_2_, t_4_, and t_24_ = 0, 1, 2, 4 and 24 h, approximately, we collected 4 mL from each reactor through the sampling septum using a spinal needle (Becton Dickinson, Madrid, Spain) and 5 mL plastic syringe (Henke Sass Wolf, Tuttlingen, Germany). The remaining particulate matter was immediately separated from dissolved ions using an Amicon Ultra-4 centrifugal filter with 3 kDa filter cut-off (product number Z740186, Merck, Darmstadt, Germany) and centrifuged at 4400× *g*, 4000 rpm, for 30 min using a Sorvall RC6+ centrifuge (Thermo Fisher Scientific, Waltham, MA, USA). Although, >95% was filtrated after 7 min, the filtration was continued for additional 23 min to ensure all was filtrated. 

It took approximately 2 min from finalizing the probe sonication and adding the particle suspension to the test reactors, until the first samples (t_0_) were collected and spinning in the centrifuge. After centrifugation, the filtrate was weighed to determine the actual sample size. To the filtered sample, 0.5 mL of 2% nitric acid (prepared in ultrapure water 18 MΩ·cm, acid obtained from Merck, Darmstadt, Germany) was added to stabilize the dissolved ions. The dissolved ionic fraction was analyzed using inductively coupled plasma-mass spectrometry (ICP-MS).

### 2.9. Inductively Coupled Plasma-Mass Spectrometry

After sampling (t_0_, t_1_, t_2_, t_4_, and t_24_), the total concentration of dissolved ions was quantified using a Thermo iCAP Q ICP-MS (Thermo Fisher Scientific, Waltham, MA, USA) equipped with an ASX-560 autosampler (Teledyne Cetac Technologies, Omaha, NE, USA). The ICP-MS was mounted with a quartz cyclonic spray chamber and a PFA-ST MicroFlow nebulizer (Thermo Fisher Scientific, Waltham, MA, USA). Throughout all ICP-MS analyses, the plasma power was 1550 W, the plasma gas flow was 14.00 L/min, the nebulizer gas flow was ~1.00 mL/min, the auxiliary gas flow was 0.80 mL/min, and the dwell time was set to 100 ms. The dilution factors and the ICP-MS parameters used to analyze the dissolved ionic fractions in PSF, and low-calcium Gamble’s solution can be found in the Appendix A.

The analyzed isotopes, internal standards and diluents for the external standards were selected based on spiking experiments. For the dissolution studies conducted in PSF, the total ion content was quantified against an external calibration curve prepared in 2% nitric acid for ZnO (NM-110, NM-111, and NM-113) and CeO_2_ (NM-212). To compensate for matrix effects, the external calibration curve for analysis of Al_2_O_3_, TiO_2_ (NM-104), SiO_2_ (NM-200), and bentonite (NM-600) was prepared in 10-, 4-, 100-, and 10-times diluted PSF, respectively, having the same dilution factor as the samples. 

Considering dissolution studies in low-calcium Gamble’s solution, the concentration of ions was quantified against an external calibration curve prepared in 2% nitric acid. ICP-MS calibration standards 1000 mg/L (SCP SCIENCE, Quebec, Canada) with trace metals ≤ 1 µg/L were used for the preparation of all external calibration curves. The internal standard was likewise prepared from 1000 mg/L ICP-MS standards and diluted to a concentration of 20 µg/L with 2% nitric acid. 

The internal standard was added to the samples online using a T-piece. Blanks and spiked samples were included in all analyses for quality control. To reduce carry-over, a rinsing procedure with 2% nitric acid was performed after all samples. During data analysis, the background measured in the blank reactor was subtracted from the test reactors. The limit of detection (LOD) of the measured ion in the undiluted sample was calculated as
(1)LOD=3·SD·DF
where *SD* is the standard deviation of ten blank samples and *DF* is the dilution factor. An overview of monitored isotopes, internal standard, and LOD in PSF and low-calcium Gamble’s solution can be found in Appendix A.

### 2.10. Determination of Initial Dissolution Rates

The dissolved ionic fractions were multiplied with the total dilution factor; corrected for adsorbed moisture, impurities, and coatings (see Section 3.1); and adjusted stoichiometrically based on the elemental and TGA-MS results to obtain the dissolved concentration of Al_2_O_3_, TiO_2_, ZnO, SiO_2_, CeO_2_, and bentonite. The total dilution factor compiles the dilution volume from acid/base titration and dilution with 0.5 mL 2% nitric acid to stabilize the sample filtrates.

Dissolution rates were calculated by determination of the reaction order using the integrate method by which SiO_2_ (NM-200), CeO_2_ (NM-212), and bentonite (NM-600) were found to follow a zero-order reaction, and the remaining materials (Al_2_O_3_, TiO_2_ (NM-104), and ZnO (NM-110, NM-111, and NM-113)) followed a mixed order or higher-order reaction. For zero-order reactions, there is a linear fit for the concentration plotted against the time. For mixed-order and higher-order reactions, the concentrations follow a non-linear regression curve as a function of time expressed by Equation (2):(2)$$C\left(t^{*}\right)=\theta_{1}-\theta_{2}\cdot \exp\left(-\theta_{3}\cdot t^{*}\right)\;\theta_{2},\;\theta_{3}>0$$
where *C(t^*^)* is the concentration as the function of time stochiometrically adjusted and corrected for impurities and moisture content, *t^*^* is the adjusted time (Equation (3)), and $\theta_{1},\;\theta_{2}$, and $\theta_{3}$ are constants. The time was adjusted to equidistant real time-points but taking the 16 min of sonication, 2 min of sampling, and 7 min of filtration into account (in total 25 min)
(3)$$t^{*}=t_{sampling}+25\;\min$$

The initial, interior, and last point dissolution rates were then determined according to Fogler (1999) [53] using the differentiation formulas:(4)$$Initial\;point:\;{\left(\frac{dC}{dt}\right)}_{t_{0}}=\frac{-3C\left(t=0\right)+4C\left(t=1\right)-C\left(t=2\right)}{2\Delta t}$$
where *C_A_* is the concentration
(5)$$Intermediate\;points:\;{\left(\frac{dC}{dt}\right)}_{t}=\left(\frac{1}{2\Delta t}\right)\left[C\left(t=t+1\right)-C\left(t=t-1\right)\right]$$
(6)$$Last\;point:\;{\left(\frac{dC}{dt}\right)}_{t_{end}}=\left(\frac{1}{2\Delta t}\right)\left[C\left(t=t_{end}-2\right)-4C\left(t=t_{end}-1\right)-3C\left(t=t_{end}\right)\right]$$

The determined numerical points are plotted as a function of *t^*^* to determine the initial dissolution rate at projected *t* = 0, ${\left(\frac{dC}{dt}\right)}_{t=0}$ by solving the regression function at time zero. The rate is then determined with the unit [mg/L/h]. The initial dissolution rate, ${\left(\frac{dC}{dt}\right)}_{t=0}$, was also provided as surface area dissolution rate considering the specific surface area (BET) by
(7)$${\left(\frac{dC\left(SSA\right)}{dt}\right)}_{t=0}={\left(\frac{dC}{dt}\right)}_{t=0}\cdot BET$$

### 2.11. Reactivity

The real-time redox reactivity was studied at time-points t_0_, t_1_, t_2_, t_4_, and t_24_ h. First, the measured redox potentials in each of the reactors were corrected for the temperature difference between *E_h_* calibration and measurement by Equation (8):(8)$$E_{h,i}=E_{h\left(batch\;reactor;i\right)}-E_{h\left(correction\right)}$$
where *E_h(batch reactor,i)_* is the redox potential measured in the batch reactor *i* with or without NMs measured in mV and *E_h(correction)_* is the function for correcting *E_h_* for the temperature difference between the standardized calibration solution and the test conditions. The reactivity (*dE_h_*) of the NM was finally determined as
(9)$$dE_{h}=E_{h\;batch\;reactor}-E_{h\;blank}$$

### 2.12. Statistics

Dissolution testing conducted in low-calcium Gamble’s solution and PSF was performed to demonstrate the use of the ATempH SBR, and the initial dissolution rates are reported as the average ± standard deviation of the three test reactors. The best fitting of the data was reported with a 95% confidence interval.

Considering the repeatability of the ATempH SBR system, a two-way repeated-measures ANOVA (analysis of variance) was used to test for independence between dissolution curves between repeats (I, II, and III). The parallelism of dissolution curves describes identical dissolution rate and behavior; however, the starting concentration can be shifted due to variations in the initial amount dissolved. Equality describes that dissolution curves are identical (same initial dissolution) and, thereby, also the initial dissolution rate. A *p*-value ≤ 0.05 was considered significant. 

In the reactivity data, the *p*-values under the null hypotheses of parallelism and equality for non-blank minus blank (*dE_h_*) reactivity curves were evaluated within repeats (I, II, and III). The hypothesis of equality was tested if the *p*-value for the null hypothesis of parallelism was greater than α = 0.05. Measurements at different time points within a replication were treated as repeated measurements with a first-order autoregressive correlation structure and a model-based covariance matrix. 

The analyses were conducted using the mixed procedure in SAS version 9.4 statistical software (SAS Institute Inc., Cary, NC, USA). In order to determine reactivity of the materials, a two-tailed Student’s t-test was conducted assuming unequal variance testing the difference between the blank (*E_h_*) and non-blank (*E_h_*) reactors. The hypothesis of reactivity was tested if the *p*-value for the null hypothesis was greater than α = 0.05. The reactivity data from testing in PSF was treated as described above.

## 3. Results and Discussion

### 3.1. Physicochemical Characteristics of the Test Materials

Table 3 summarizes the physicochemical characteristics of the test materials applied in the study. The results show the essential parameters, including the adsorbed moisture, coating, and impurities, that need to be considered when calculating the material solubility and dissolution rate.

Al_2_O_3_ was identified as ɣ-aluminum oxide by the supplier [54], which was supported in a study conducted by Krause et al. (2018) [61]. However, several unidentified XRD peaks were observed in the spectrum shown by Krause et al. (2018). In this study, we confirmed the presence of ɣ-Al_2_O_3_ and identified the minor crystalline phase as aluminum oxyhydroxide (boehmite) and an additional phase that may be barentsite (Appendix A). The sample potentially also contains a significant amount of amorphous material. The TGA-MS analysis revealed 3.77 wt% moisture content and no mass-losses at higher temperatures. 

The technical report from IoLiTec did not report any moisture in the product. No loss on ignition was found, which would be expected in the presence of boehmite. For determination of the specific surface-area dissolution rate, we used the supplier’s data (200 m^2^/g).

According to the supplier, TiO_2_ (NM-104) is rutile and is coated with Al_2_O_3_ with 6 wt% and 2 wt% glycerin. The WDXRF analysis showed that the Al_2_O_3_ content was 6.08 wt%. The TGA-MS results showed a 1.50 wt% moisture content and 3.11 wt% inorganic coating. These values are in good agreement with recent data in Clausen et al. (2019) who reported a moisture content of 1.49 wt% and an inorganic coating of 3.17 wt% [50].

ZnO (NM-110) is an un-coated zincite. The results from our WDXRF analysis consistently showed unusual high concentrations of Ti (1.20 wt% TiO_2_). A previous study reported 1.1 wt% Na (as Na_2_O), NANoREG database, which we here mainly ascribe to influence of peak interference. The TGA-MS analysis showed 0.28% moisture. The TGA analysis of NM-110 also showed an episodic mass-loss of between 220 and 260 °C, which was not previously reported by Singh et al. (2011) [62].

ZnO (NM-111) is a zincite and, according to the supplier, coated with triethoxycaprylsilane. The XRD analysis showed a purity of 97.62 wt% ZnO and smaller fractions of impurities primarily assigned to SiO_2_ and TiO_2_. TGA analysis of NM-111 showed no moisture content but an episodic mass-loss of 1.59 wt% between 200 and 500 °C, which is slightly lower than the 2.1 wt% observed in previous analysis Clausen et al. (2019) [50].

ZnO (NM-113) is, according to the supplier, an uncoated zincite. WDXRF analysis showed that the sample was relatively pure with 99.17 wt% ZnO. The moisture content was low, with 0.69 wt%. The LOI was 0.20 wt%, and the entire mass was lost episodically between 240 and 260 °C.

SiO_2_ (NM-200) is synthetic amorphous silica (SAS) [57]. The WDXRF data showed a silica content of only 82.08 wt% and Na_2_O, Cl, and SO_3_ due to the presence of sulfate and salt impurities. Rasmussen et al. (2013) reported the main impurities as Na_2_SO_4,_ and γ-AlO(OH) detected using XRD. Aluminum and titanium were also observed, corresponding to impurities of 0.94 wt% Al_2_O_3_ and 1.05 wt% TiO_2_, respectively [63]. TGA-MS analysis showed a moisture content of 5.08 ± 0.12 wt% and an LOI of 3.8 ± 0.13 wt%. The mass-loss in LOI generally occurs gradually between 100 to 800 °C, with most losses reached at approximately 600 °C. Part of the mass loss in LOI is due to the decomposition of the impurity phases.

CeO_2_ (NM-212) is a cerianite that showed a purity of 97.90 wt%. The moisture content was relatively low (0.13 wt%), and the LOI was 0.71 wt%, which composed between 108–800 °C.

Bentonite (NM-600) is considered a nanoclay, and the fundamental physicochemical properties were previously reported by OECD [59]. X-ray diffraction analysis showed impurity of crystalline silica (quartz and cristobalite) (Appendix A). The WDXRF data showed a content of 17.57 wt% Al_2_O_3_ and 53.05 wt% SiO_2_, which are considered the main components of bentonite (NM-600). The moisture content was 6.63 wt% ascribed to interlayer water molecules, and the 5.29 wt% mass loss in LOI occurred between 110 and 360 °C. The presented chemical composition was in good correlation with a study from Pereira et al. (2012) [64]. The mineral chemical structure composition was established by assuming all Fe as divalent and considering K, P, S, and Cl as impurities in addition to 1.9 wt% silica and 0.7 wt% Na_2_O:
(Ca_0.09_, Na_0.58_)_0.7_(Al_2.92_, M^1^_0.68_)_3.6_[(Si_7.7_, Al_0.2_, M^2^_0.08_)_8.0_O_20_](OH)_4_
where M^1^ is Fe^2+^_0.26_, Ga^3+^_0.004_, Mg^2+^_0.41_, Zn^2+^_0.002_, Mn^2+^_0.001_, and Cu^2+^_0.001_ and M^2^ is Nb^5+^_0.00017_, Ti^4+^_0.074_, and Zr^4+^_0.001_.

### 3.2. Repeatability and Robustness

In the context of this work, we define repeatability as the variation between repeats (I, II, and III). The variation includes contributions from NM dispersion (weighing of the powder, addition of dispersion liquids, and sonication procedure), addition of dispersion to the ATempH SBR system, sampling from the ATempH SBR system (t_0_, t_1_, t_2_, t_4_, and t_24_), and to a lesser degree ICP-MS analysis. The most critical parameters influencing the repeatability were identified as preparation and addition of the dispersion. 

We here define robustness as the ATempH SBR system’s resistance to the influence of technical system performance (constant air flow (CO_2_ and O_2_), temperature, and pH) and use of different simulated lung media, and different test materials. Table 4 provides an overview of the system performance in the two test media. All measured parameters showed minimal variations throughout 24 h of testing, documenting the good test control of the system.

### 3.3. Particle Dispersion

The mean hydrodynamic size (Z_ave_), polydispersive index (PDI) and zeta potential (ζ_pot_) of each batch dispersion were measured and used as quality control of the particle dispersions to report under which conditions the dissolution testing was conducted. The three particle dispersions (I, II, and III) used for the validation studies in low-calcium Gamble’s solution dissolution are found in Table 5, and the hydrodynamic size spectra are in the Appendix A.

The three replicates of Al_2_O_3_ dispersions showed good comparability in terms of Z_ave_, PDI and ζ_pot_. The relatively high negative ζ_pot_ values indicated that the dispersions were stabilized by charge. Particle dispersions with a ζ_pot_ < −30 mV or > +30 mV are considered as stable dispersions [52,65]. The addition of 0.05% BSA further supported the stabilization of the suspension as described by Hartmann et al. (2015) [66]. The low PDI (< 0.3) showed that Al_2_O_3_ formed monodispersive agglomerates in the suspensions.

ZnO (NM-110 and NM-111) showed comparable Z_ave_ and negative ζ_pot_s for the replicates and between the two materials. The initial particle sizes of the two materials are likewise comparable (Table 3). One could expect that the organic triethoxycaprylsilane coating of NM-111 would affect the dispersibility; however, this was not observed. The slightly negative ζ_pot_ (−10 mV) classifies the material as neutrally charged [67], though (sterically) stabilized by the addition of 0.05% BSA. 

The PDI’s indicated that both ZnO (NM-110 and NM-111) dispersions have a relatively narrow size distribution. The ZnO (NM-113) showed a non-systematic variation in the Z_ave_ between the replicates (I, II, and III). ZnO (NM-113) has a larger primary particle size than ZnO (NM-110), which may explain the formation of larger agglomerates. Despite the differences in Z_ave_, the ζ_pot_s of the replicates were comparable ~−7 mV and showed low PDIs.

Agglomeration occurred to a high extent for both TiO_2_ (NM-104) and SiO_2_ (NM-200). In both cases, the Z_ave_s were greater than previously reported [51,68]. The repeatability of the TiO_2_ (NM-104) dispersion was relatively poor, as one dispersion showed a significantly lower Z_ave_ (replicate II) compared with replicate I and III. Previously, vial-to-vial and within-vial variation were recognized (data not published), which may affect the dispersion quality and potentially the dissolution behavior of the materials. TiO_2_ (NM-104) showed a zeta potential close to zero, and it was therefore primarily stabilized by the 0.05% BSA. Despite the large Z_ave_ of SiO_2_ (NM-200) and high PDI (~1), the material showed high negative ζ_pot_s. No visual indication of rapid sedimentation was observed for TiO_2_ (NM-104) and SiO_2_ (NM-200) despite the large agglomerates.

In the case of CeO_2_ (NM-212), the three dispersions were shown to be repeatable in terms of Z_ave_. The high positive ζ_pot_s and low PDI (<0.3) further indicated the dispersions to be stable and with relatively narrow size distributions. Bentonite (NM-600) dispersions were comparable in Z_ave_, though the dispersions showed a relatively broad size distribution (PDI > 0.3). The zeta potentials of the dispersions were highly negative (<−30 mV). Dispersions made for dissolution testing in PSF were likewise dispersed in 0.05% BSA. The results are found in Table 6, and size distributions are found in the Appendix A.

The TiO_2_ (NM-104) dispersion used for testing in PSF showed a significantly lower Z_ave_ than the replicates used for testing in low-calcium Gamble’s solution. The ζ_pot_ and PDI were comparable to the results of TiO_2_ (NM-104) in Table 5. SiO_2_ (NM-200) showed poor dispersibility as reported in Table 5.

The ZnO materials (NM-110, NM-111, and NM-113) demonstrated comparable dispersion parameters as seen in the previous dispersions for experiments in Gambles solution (Table 5). This was also the case of CeO_2_ (NM-212), though the ζ_pot_ was significantly lower and close to zero for this dispersion. Al_2_O_3_ and bentonite (NM-600) also showed comparable dispersion quality as previously described in for testing conducted in low-calcium Gamble’s solution; Table 5.

### 3.4. Reactivity

The experimental conditions during 24 h were measured to document the reactivity during dissolution testing. The measured redox potential values in both low-calcium Gamble’s solution (Appendix A) and PSF (Appendix A) were in good agreement with the reported values for biological compartments described by Plumlee and Ziegler (2003) [48]. Statistically, the redox potential was tested against the reference reactor for parallelism and equality of the measured values to investigate the repeatability of the ATempH system. The *p*-values for testing conducted in low-calcium Gamble’s solution are found in Table 7. All *E_h_* values, statistics, and *dE_h_* values are found in the Appendix A.

Al_2_O_3_, TiO_2_ (NM-104), ZnO (NM-110 and NM-113), and SiO_2_ (NM-200) were shown to be parallel and equal across the three reactors containing NMs, thereby, showing repeatability in the ATempH system. The statistical tests of ZnO (NM-111), CeO_2_ (NM-212), and bentonite (NM-600) were found to be not fully parallel. This appears to be due to larger differences between reactors (replicate 1, 2, and 3) rather than between the overall variation in the redox potentials.

To examine the reactivity of the materials in both low-calcium Gamble’s solution and PSF during the 24 h, a Student’s t-test was performed at all time points, testing the blank reactor against the three reactors containing NMs. The results are found in Appendix A. All eight NMs showed to be reactive in both low-calcium Gamble’s solution and PSF. 

In general, the redox potential, *E_h_* (Equation (8)), was higher in PSF during the entire 24 h of testing as compared to the redox potential in low-calcium Gamble’s solution, resulting from the differences in medium composition and pH [48]. ZnO (NM-110 and NM-111) and bentonite (NM-600) showed higher *dE_h_* values in PSF compared to that in low-calcium Gamble’s solution. Al_2_O_3_, TiO_2_ (NM-104), SiO_2_ (NM-200), and CeO_2_ (NM-212) showed comparable *dE_h_* values in both low-calcium Gamble’s solution and PSF. There is clearly an influence from the material on the redox potential, which will be studied in future work (Appendix A).

### 3.5. Repeatability of the ATempH SBR System

The repeatability of the SBR system was tested in low-calcium Gamble’s solution by three repeated measurements (I, II, and III). Table 8 below provides an overview of the determined initial dissolution rates for each of the eight test materials.

For Al_2_O_3_, the dissolution profile (Figure 2) followed a non-linear regression curve as described in Equation (2). There was no significant difference found between the three repeats. The dissolution profiles of the three repeats were found to be parallel (*p*-value: 0.5277) and equal (*p*-value: 0.4137), statistically showing identical dissolution behavior and initial dissolution rates across the three repeats.

TiO_2_ (NM-104) showed no dissolution of titanium <LOD. However, the aluminum coating was found to dissolve during 24 h of testing in the ATempH SBR following a non-linear regression curve (Equation (2)). The three repeats (I, II, and III) of TiO_2_ (NM-104) showed slightly different dissolution behaviors (*p*-value: 0.0074). Repeats I and III created the largest agglomerates and showed comparable dispersibility. Repeat II had a smaller hydrodynamic size after probe sonication. However, repeats I and II were highly similar in terms of dissolution; Figure 3.

The observed variations may be due to variations in the coating quality of the material. Uneven coating (within-vial or vial-to-vial variation) will influence the dissolution and repeatability of the ATempH SBR system. One could think the material used in repeat (I) contained less Al_2_O_3_ coating. The authors acknowledge the statistical differences but do not, however, expect the variation to influence the overall understanding of the dissolution of TiO_2_ (NM-104) in low-calcium Gamble’s solution.

The dissolution of the three ZnO (NM-110, NM-111, and NM-113) materials all followed a non-linear fit. Thereby, the kinetics were determined by the numerical differential method. For ZnO (NM-110), the three repeats were parallel (*p*-value: 0.6578), therefore, having the same initial dissolution rate across the three repeats. The equality test showed a significant difference (*p*-value < 0.0001), as expected from Figure 4. Repeat I had a higher offset than repeats II and III. As the dispersibility quality parameters across the three repeats are considered identical; minor vial-to-vial inhomogeneity in the ZnO may have caused the change in offset.

The three repeats of ZnO (NM-111) were found to be parallel (*p*-value: 0.0627), though not equal (*p*-value: 0.0051), Figure 5. The organic triethoxycaprylsilane coating could potentially affect the solubility of Zn^2+^-ions as the coating has to dissolve or disintegrate before ZnO can dissolve. Therefore, the dissolution and/or disintegration of the organic coating is an essential factor affecting the dissolution of Zn^2+^. Inhomogeneous coating with the organic triethoxycaprylsilane could also potentially affect the dissolution of Zn^2+^. The coating might explain why variations between the three repeats (I, II, and III) were observed.

In the case with ZnO (NM-113), the three repeats were found to be parallel (*p*-value: 0.4210) and equal (*p*-value: 0.1727), Figure 6. Therefore, no significant variation between the three repeats was found. Comparing the three ZnO (NM-110, NM-111, and NM-113) materials, the ATempH SBR system demonstrated the ability to determine differences in dissolution behavior of the three ZnO materials. ZnO (NM-110) showed the fastest dissolution rate of 0.112 ± 0.082 cm^2^/L/s) followed by ZnO (NM-111) 0.074 ± 0.033 cm^2^/L/s, and ZnO (NM-113) 0.036 ± 8.28 × 10^−3^ cm^2^/L/s.

The differences were considered to be a result of the size (and specific surface area) difference across the materials, but they did not reach similar values in the surface-area dissolution rate. Singh et al. (2011) evaluated the size of ZnO and found a primary particle size of 20–250 nm for ZnO (NM-110), approximately 90% by number 20–200 nm for ZnO (NM-111), and 40–500 nm for ZnO (NM-113) determined using TEM [62].

ZnO (NM-110 and NM-111) showed comparable primary particle sizes with different dissolution rates; however, the organic coating of ZnO (NM-111) potentially lowered the initial dissolution rate. ZnO (NM-113) had the largest primary particle size and showed the lowest initial dissolution rate of the three ZnO materials in this study.

The dissolution of SiO_2_ (NM-200) followed a linear regression measured between 0–24 h, Figure 7. The dissolution was a zero-order reaction [69]. As the kinetic reaction of SiO_2_ (NM-200) follows a zero-order reaction, the dissolution rate was determined to be the slope of the linear plot of concentration (mg/L) as a function of time. Expectedly, SiO_2_ (NM-200) was very soluble in low-calcium Gamble’s solution due to the neutral pH (7.4). SiO_2_ (NM-200) demonstrated the fastest initial dissolution rate of the tested materials. 

The initial rates were found to be significantly different (*p*-value < 0.0001) between the three repeats. The suspension used for repeat III showed the smallest hydrodynamic size after probe sonication. Intuitively, a smaller hydrodynamic size would result in a faster dissolution; however, this was not observed.

As previously reported, the chemical composition and phase composition of bentonite (NM-600) are potentially complex, and dissolution will result in release of a variety of minor and trace element ions. In terms of dissolution, only aluminum and silicon were studied as representative of the core in the crystalline structure, Figure 8. No dissolution of aluminum above LOD was found. The release of silicon followed a linear fit, indicating that the release of silicon was a zero-order kinetic reaction. However, the three repeats showed significantly different dissolution behaviors (*p*-value: 0.0052); the second repeat (II) showed no dissolution of silicon. 

The atomic Si-Al-Si sandwich layer of the montmorillonite in the bentonite (NM-600) could potentially hamper the release of silicon. However, we observed that silicon was also present in quartz and cristobalite impurities, and the potential presence of amorphous silica is currently unknown. In future work, multi-elemental analysis of the bentonite (NM-600) dissolution behavior coupled with detailed electron microscopy analysis might provide a more detailed understanding of the dissolution behavior of bentonite.

### 3.6. Dissolution in Phagolysosomal Fluid

The testing conducted in PSF was used for further demonstration of the ATempH SRB system, depicted in Appendix A. Table 9 provides an overview of the calculated initial dissolution rates of the eight test materials in PSF.

Compared with dissolution rates in low-calcium Gamble’s solution, Al_2_O_3_ and SiO_2_ (NM-200) showed a slower dissolution in PSF. Al_2_O_3_ followed a non-linear fit, and SiO_2_ (NM-200) showed the best fit with linear regression. Therefore, the materials follow the same type of dissolution as in low-calcium Gamble’s solution, however, at a slower rate. Expectedly, the dissolution was lower of SiO_2_ (NM-200) in a medium with an acidic pH (4.5).

TiO_2_ (NM-104) showed no dissolution of titanium <LOD, but the inorganic aluminum coating dissolved at a comparable dissolution rate as found in low-calcium Gamble’s solution. The dissolution kinetics again followed a non-linear fit. Comparably, Koltermann-Jülly et al. (2018) studied the dissolution of TiO_2_ (NM-104 and NM-105) in PSF using a flow-through system. The authors found no dissolution of titanium above LOD but did not investigate the dissolution of aluminum of TiO_2_ (NM-104) [31].

As for bentonite (NM-600), only the dissolution of silicon was detected. As mentioned above, quartz and tridymite in addition to the potential presence of amorphous silica may also contribute to the measured silicon release. The lack of aluminum dissolving may indicate that bentonite (NM-600) itself may not dissolve. The dissolution followed a zero-order reaction and showed comparable dissolution rates as found in low-calcium Gamble’s solution.

The low pH of PSF favored the dissolution of CeO_2_ (NM-212). Within the first four hours, the dissolution of CeO_2_ followed a linear fit and was, thereby, a zero-order reaction. After 24 h, the materials appeared to reach the solubility limit, and the time-point t_24_ h was therefore excluded for determination of the initial dissolution rate. In contrast, Koltermann-Jülly et al. (2018) showed no dissolution of CeO_2_ in PSF measured over 24 and 168 h in a flow-through system [31].

In general, the ATempH SBR system showed high repeatability, as relatively small standard variations were found across the three replicates (test reactors) for all eight materials demonstrating the ATempH SBR system performed identically within 24 h of testing. The ATempH SBR system generally demonstrated high intra-laboratory repeatability within the repeated testing of the materials. Though, the aluminum coating of TiO_2_ (NM-104), SiO_2_ (NM-200), and bentonite (NM-600) statistically showed significant differences across the replicates (I, II, and III). 

As previously discussed, uneven coatings of TiO_2_ (NM-104) and the molecular structure of bentonite (NM-600) possibly influenced the dissolution of Al^3+^ and Si^2+^ ions, respectively. In the case of SiO_2_ (NM-200), minor variations within the vial could have influenced the repeatability, as previous within-vial and vial-to-vial variations have been recognized for this material. Variation at this level was, therefore, accepted. Al_2_O_3_, ZnO (NM-110, NM-111, and NM-113), and CeO_2_ (NM-212) statistically showed no variations across the dissolution profiles and rates in all three repeats. The performance of the ATempH was, therefore, independent of time. This validation was limited to intra-laboratory validation, as the presented ATempH SBR is the first of its kind.

In this study, we presented the measured dissolution rates with uncertainties, which has not been the standard procedure from previous dissolution studies of NMs [31,45,70]. The percentage deviation from the average determined dissolution rate was found as low as 3.1% (SiO_2_ (NM-200), repeat I) and as high as 94% (ZnO (NM-111), repeat II). However, the average deviation was approximately 3–55% in low-calcium Gamble’s solution. The performance in PSF was found with percentage deviations from 0.3% to 22%. The authors acknowledge that further inter- and intra-laboratory testing are needed to understand the background for observed differences, which may also be linked to differences in the different materials’ homogeneity and dissolution behavior as well as the role of predisperison quality. 

The ATempH SBR showed robustness as different lung media with different pH and salt composition could be applied, that NMs with low and high solubility could be tested in both types of media, and the ATempH SBR system technical performance was identical across testing. The use of the ATempH SBR system is not limited to nanoclays and metal-oxide (nano)materials—the dissolution of carbon-based and pure metal-based materials, etc. can also be conducted. However, we chose well-characterized coated and uncoated OECD materials with slow, partial, and high dissolution rates for validation and demonstration of the ATempH SBR system.

Preparing the ATempH SBR system for dissolution testing requires approximately 2–3 h. The relatively time-consuming preparation of the ATempH SBR system recompense with simultaneously triplicate dissolution testing of an NM conducted under exactly equal experimental conditions. Further, the ATempH SBR system provides tight control of the pH, temperature, gas flow, and composition, which is required to gain a better understanding of the actual dissolution properties in biological compartments. A drawback of the ATempH SBR system is the need for a pre-dispersion step and the importance of precise dosing of NM dispersions. 

The dispersions were made following the NANOGENOTOX protocol, which presents a harmonized dispersion protocol of NMs [51]. A harmonized protocol allows direct comparison of the dispersed materials. Despite the user-friendly ATempH SBR system, it was impossible to incorporate a preliminary quality test of the NM dispersions before the suspension was added to the test reactors. Instead, the quality was determined while the dissolution test was running. Poor dispersions would therefore only be recognized after the dissolution test was run. Future work is needed to investigate the role of the dispersibility of test materials to understand the importance of this parameter (1) for accurate dosing and (2) on dissolution rates.

## 4. Conclusions

In this study, we described a new dissolution system for studying dissolution behavior of NMs. The ATempH SBR system was capable of controlling the temperature, pH, gas flow, and composition during testing in order to lock the conditions relevant for human lungs. Further, the ATempH SBR dissolution system demonstrated the potential of measuring the redox potential during 24 h of dissolution. The intra-laboratory repeatability of the new ATempH SBR system was tested in triplicate on eight different NMs in low-calcium Gamble’s solution. 

The system showed high repeatability for Al_2_O_3_, ZnO (NM-110, NM-111, and NM-113), and CeO_2_. Significant variations were found for TiO_2_ (NM-104), SiO_2_ (NM-200), and bentonite (NM-600). Despite the variations for three of the materials, the ATempH SBR system was considered robust overall and allowed the generation of repeatable results. As a demonstration of the potential of the system, the eight NMs were tested in PSF. The different pH value of PSF (pH = 4.5) resulted in different dissolution behaviors of the eight materials. To increase the predictability between dissolution and toxicological studies, it is essential to mimic human conditions during the dissolution testing. With the ATempH SBR system, it is possible to establish such experimental conditions relatable to biological compartments.

## Figures and Tables

**Figure 1 nanomaterials-12-00517-f001:**
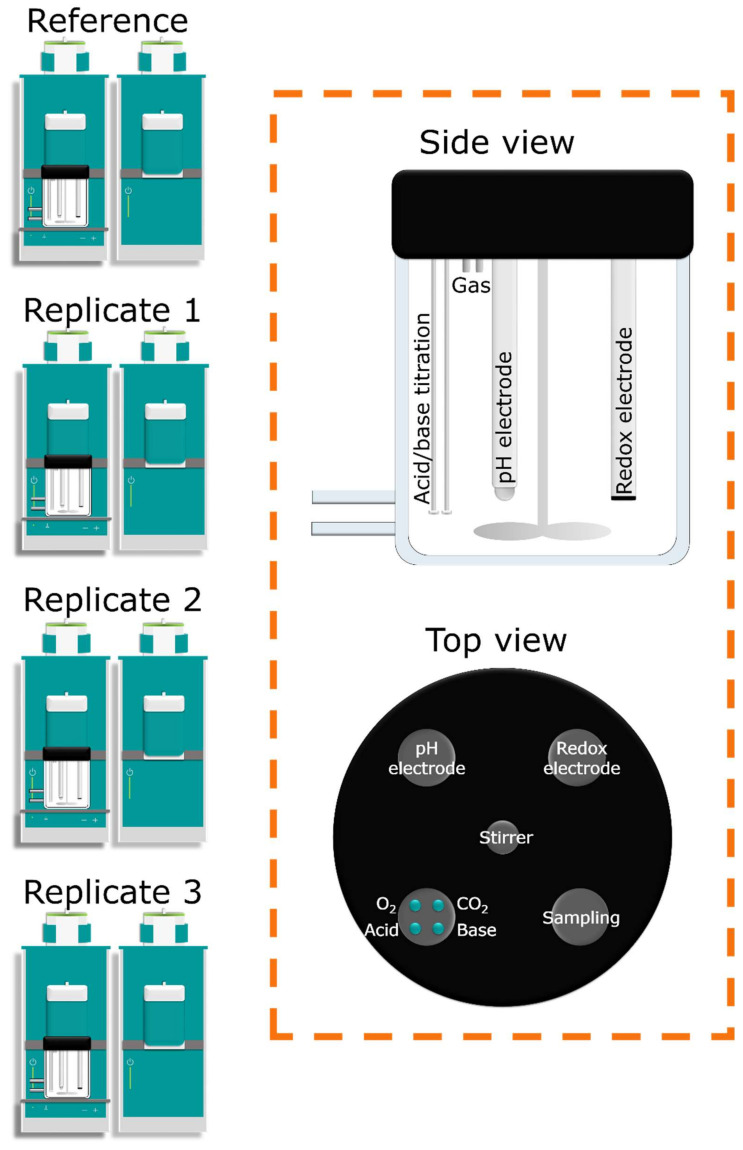
Illustration of the atmosphere–temperature-pH-controlled stirred batch reactor (ATempH SBR) system. Features of the ATempH SBR include stirring, pH regulation, and measurement of the redox potential, gas flow, and composition. The ATempH SBR consists of four units, one used as a reference and three (replicates 1–3) used for dissolution testing in one repeat.

**Figure 2 nanomaterials-12-00517-f002:**
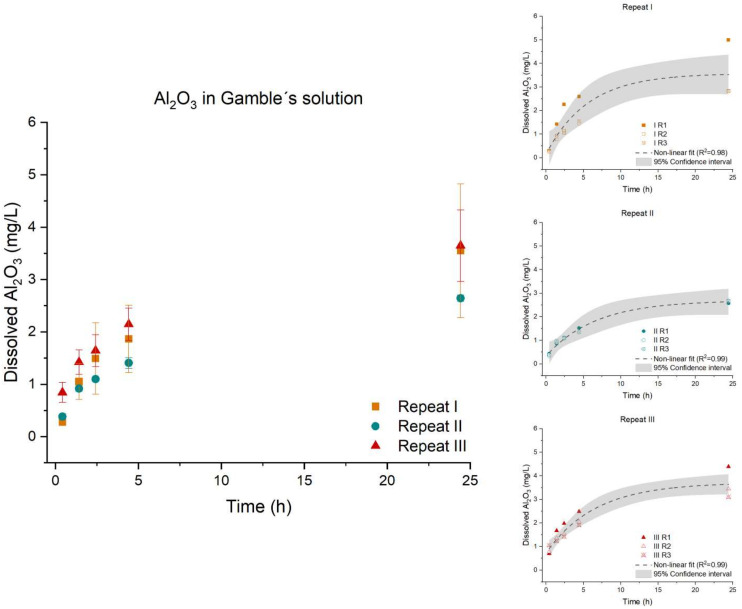
Left: Dissolution profile of Al_2_O_3_. The test was conducted in low-calcium Gamble’s solution in triplicate (*n* = 3) with three repeated tests (I (■), II (●), and III (▲)). Right: The batch reactor variation for each repeat is shown, including a 95% confidence interval.

**Figure 3 nanomaterials-12-00517-f003:**
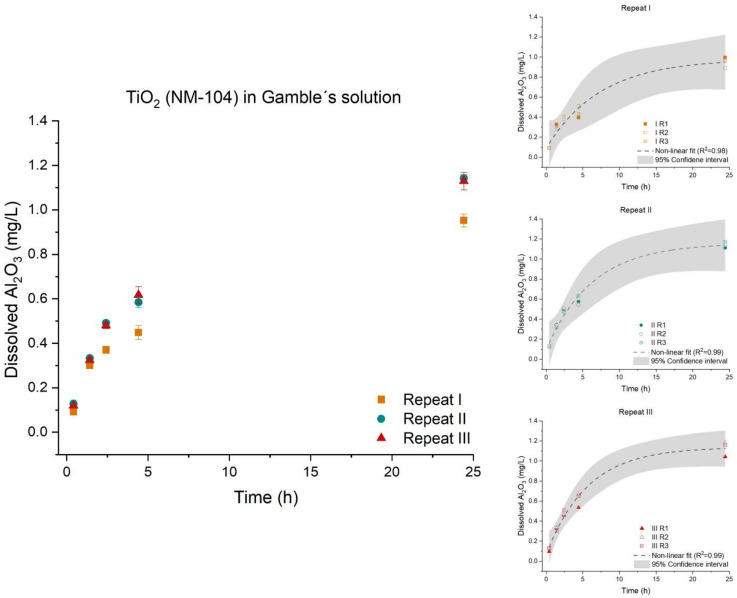
Left: Dissolution profile of the dissolved inorganic Al_2_O_3_ coating of TiO_2_ (NM-104). The solubility of Ti was below the limit of detection. The test was conducted in low-calcium Gamble’s solution in triplicate (*n* = 3) with three repeated tests (I (■), II (●), and III (▲)). Right: The batch reactor variation for each repeat is shown, including a 95% confidence interval.

**Figure 4 nanomaterials-12-00517-f004:**
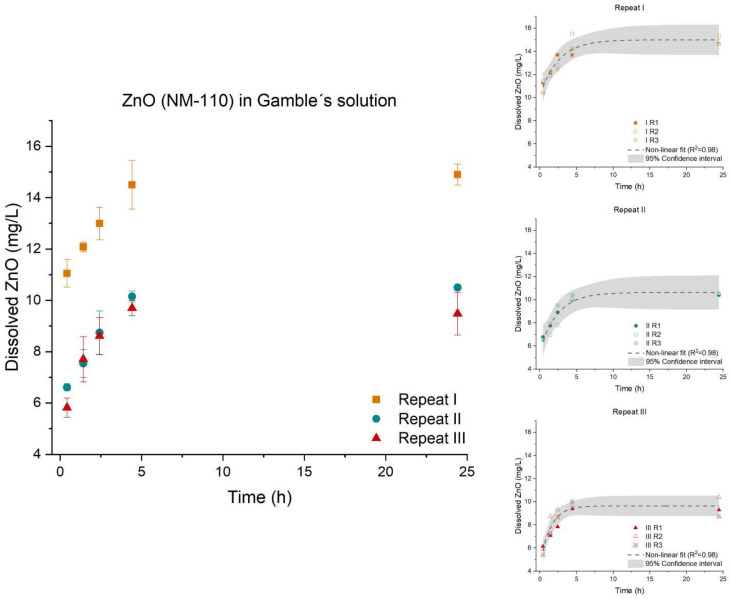
Left: Dissolution profile of ZnO (NM-110). The test was conducted in low-calcium Gamble’s solution in triplicate (*n* = 3) with three repeated tests (I (■), II (●), and III (▲)). Right: The batch reactor variation for each repeat is shown, including a 95% confidence interval.

**Figure 5 nanomaterials-12-00517-f005:**
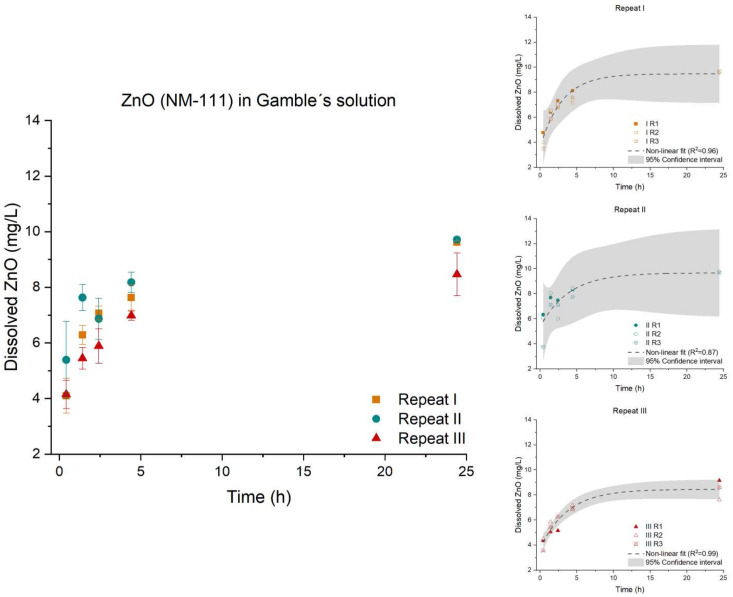
Left: Dissolution profile of ZnO (NM-111). The test was conducted in low-calcium Gamble’s solution in triplicate (*n* = 3) with three repeated tests (I (■), II (●), and III (▲)). Right: The batch reactor variation for each repeat is shown, including a 95% confidence interval.

**Figure 6 nanomaterials-12-00517-f006:**
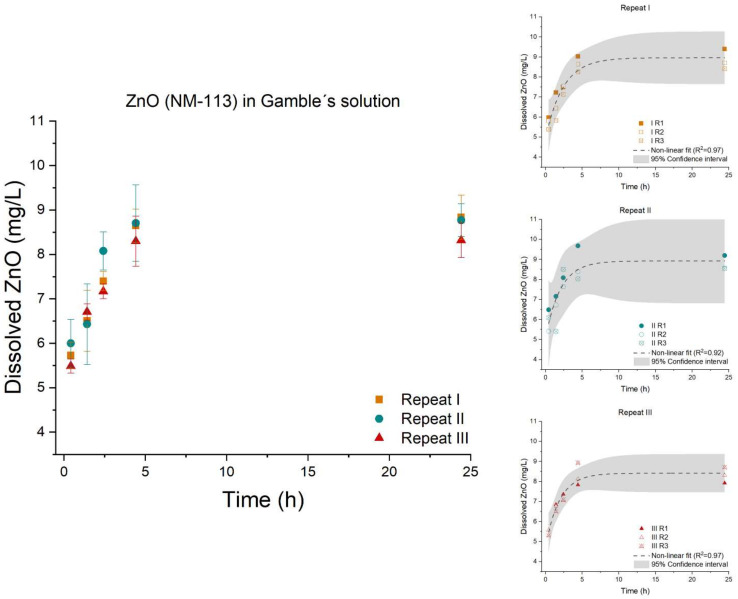
Left: Dissolution profile of ZnO (NM-113). The test was conducted in low-calcium Gamble’s solution in triplicate (*n* = 3) with three repeated tests (I (■), II (●), and III (▲)). Right: The batch reactor variation for each repeat is shown, including a 95% confidence interval.

**Figure 7 nanomaterials-12-00517-f007:**
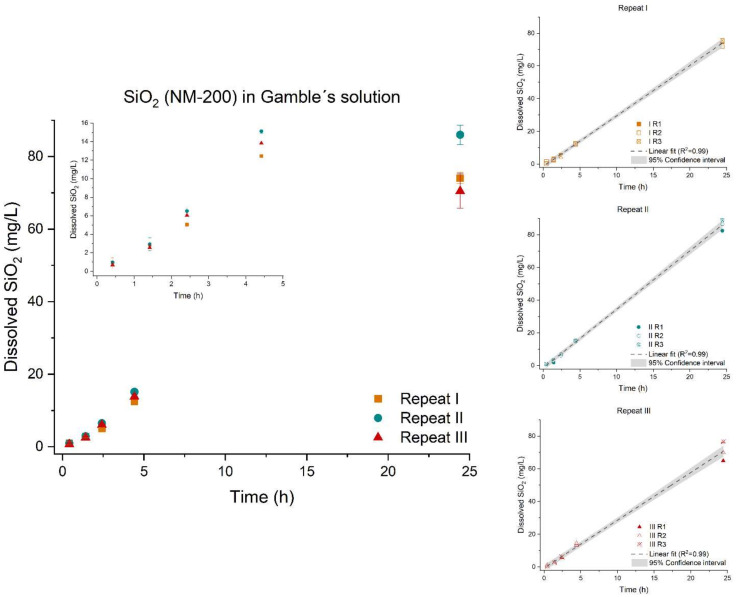
Left: Dissolution profile of SiO_2_ (NM-200). The test was conducted in low-calcium Gamble’s solution in triplicate (*n* = 3) with three repeated tests (I (■), II (●), and III (▲)). Right: The batch reactor variation for each repeat is shown, including a 95% confidence interval.

**Figure 8 nanomaterials-12-00517-f008:**
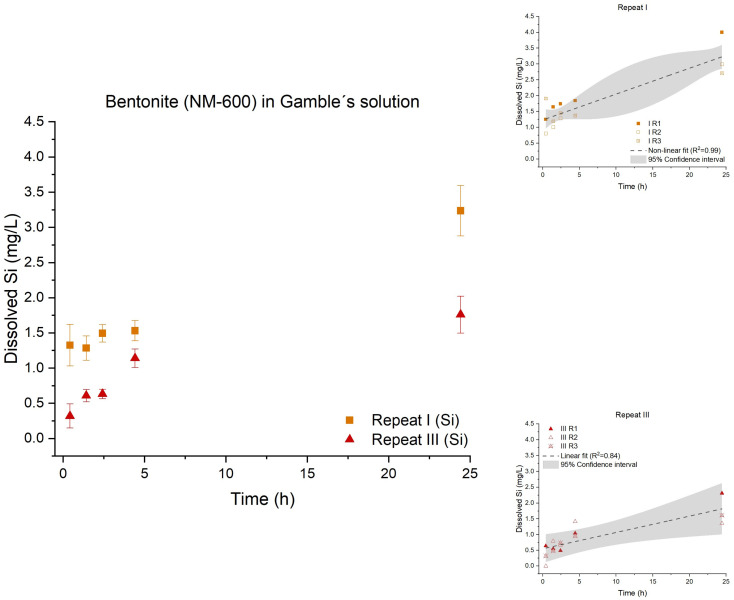
Left: Dissolution profile of silicon (Si) from nanoclay bentonite (NM-600). No dissolution of silicon was detected for the second repeat (II). The test was conducted in low-calcium Gamble’s solution in triplicate (*n* = 3) with three repeated tests (I (■) and III (▲)). Right: The batch reactor variation for each repeat is shown, including a 95% confidence interval.

**Table 1 nanomaterials-12-00517-t001:** Composition of the phagolysosomal simulant fluid. Adapted from Ref. [43].

Component	Chemical Formula	Concentration [mg/L]
Sodium phosphate dibasic anhydrous	Na_2_HPO_4_	142
Sodium chloride	NaCl	6650
Sodium sulfate anhydrous	Na_2_SO_4_	71
Calcium chloride dihydrate	CaCl_2_·2H_2_O	29
Glycine	H_2_NCH_2_CO_2_H	450
Potassium hydrogen phthalate	(1-(HO_2_C)-2-(CO_2_K)-C_6_H_4_)	4085
Alkylbenzyldimethylammonium chloride	-	50

**Table 2 nanomaterials-12-00517-t002:** Composition of the low-calcium Gamble’s solution. Adapted from Ref. [27].

Component	Chemical Formula	Concentration [mg/L]
Sodium chloride	NaCl	6600
Sodium bicarbonate	NaHCO_3_	2703
Calcium chloride	CaCl_2_	22
Sodium phosphate dibasic dodecahydrate	Na_2_HPO_4_·12H_2_O	358
Sodium sulfate anhydrous	Na_2_SO_4_	79
Magnesium chloride hexahydrate	MgCl·6H_2_O	212
Glycine	H_2_NCH_2_CO_2_H	118
Sodium citrate dihydrate	Na_3_C_6_H_5_O_7_·2H_2_O	153
Sodium tartrate dihydrate	Na_2_C_4_H_4_O_6_·2H_2_O	180
Sodium pyruvate	C_3_H_3_NaO_3_	172
Sodium lactate	C_3_H_3_NaO_3_	175

**Table 3 nanomaterials-12-00517-t003:** Key physicochemical characteristics of the test materials.

Nanomaterial	Al_2_O_3_	TiO_2_ NM-104	ZnO NM-110	ZnO NM-111	ZnO NM-113	SiO_2_ NM-200	CeO_2_ NM-212	Bentonite NM-600
Phase	ɣ-Al_2_O_3_	Rutile	Zincite	Zincite	Zincite	Synthetic amorphous silica	Cerianite	Montmorillonite, nanoclay
Specific surface area (SSA) [m^2^/g]	<200 ^a^	58.5 ± 46.3 ^b^	12.4 ± 0.6 ^c^	15.1 ± 0.6 ^c^	6.21 ± 0.4 ^c^	342 ± 36 ^d^	27.2 ± 0.9 ^e^	51.9 ± 1.6 ^f^
Inorganic coating	-	Al_2_O_3_	-	-	-	-	-	-
Organic coating	-	Glycerin ^g^	-	Triethoxy- caprylsilane	-	-	-	-
Na_2_O [%]		0.12	-	-	-	1.65	-	2.68
Al_2_O_3_ [%]	102.17	6.08	-	-	-	0.94	0.75	17.57
SiO_2_ [%]	0.03	0.13	-	0.73	-	82.08	0.14	53.05
P_2_O_5_ [%]	0.0093	-	-	-	-	0.025	-	0.013
SO_3_ [%]	0.084	0.65	-	-	0.05	1.83	0.39	0.59
Cl [%]	0.014	0.03	-	-	0.02	0.11	0.13	0.14
K_2_O [%]		-	-	-	-	0.03	-	0.06
CaO [%]		0.05	-	-	-	0.07	-	0.57
TiO_2_ [%]		91.44	1.24	0.34	0.20	1.05	-	0.65
Fe_2_O_3_ [%]	0.0034	0.01	0.01	0.007	-	0.04	0.08	4.62
Ga_2_O_3_ [%]		-	-	-	-	-	-	0.0044
CoO [%]		-	-	-	-	-	0.03	-
NiO [%]		-	0.007	0.007	-	0.0036	-	-
CuO [%]	0.0032	0.006	0.04	0.03	0.04	0.01	0.04	0.0053
ZnO [%]		-	97.62	97.86	99.17	0.01	0.09	0.019
MgO [%]		-	-	-	-	0.007	0.09	1.80
MnO [%]		-	-	-	-	-	-	0.0066
ZrO_2_ [%]		0.003	-	-	-	0.0067	-	0.017
MoO_3_ [%]		-	-	-	-	0.0057	-	-
Nb_2_O_5_ [%]		0.02	-	-	-	-	-	0.0025
CeO_2_ [%]		-	-	-	-	-	97.70	-
Adsorbed moisture [%]	3.77	1.50 ± 0.10	0.28 ± 0.11	ND	0.69	5.08 ± 0.12	0.13	6.63
	(*n* = 2)	(*n* = 3)	(*n* = 3)	(*n* = 3)	(*n* = 1)	(*n* = 3)	(*n* = 1)	(*n* = 1)
LOI ^h^ [%]	ND	3.11 ± 0.12 ^j^	0.59 ± 0.27 ^i,j^	1.59 ± 0.07 ^j^	0.20 ^j^	3.80 ± 0.13	0.71	5.29 ^k^
	(*n* = 1)	(*n* = 3)	(*n* = 3)	(*n* = 3)	(*n* = 1)	(*n* = 3)	(*n* = 1)	(*n* = 1)
Total [%]	106.09	103.13	99.77	100.57	100.37	96.75	100.28	93.72

^a^ Technical report Ionic Liquids Technologies GmbH (2019) [54]. ^b^ De Temmerman et al., NANoGENOTOX deliverable 4.2 (2012) [55]. ^c^ OECD (2015) [56]. ^d^ Rasmussen et al. (2013), JRC Repository [57]. ^e^ Singh et al. (2014), JRC repository [58]. ^f^ OECD (2015) [59]. ^g^ OECD (2016) [60]. ^h^ Loss on Ignition is in this study defined as the mass-loss obtained between the temperature used for determination of water-loss (below 100–110 °C), and the maximum temperature in the TGA analysis performed. ^i^ Small mass-gain is observed above 410 °C. ^j^ Episodic mass-loss event ascribed to coating or impurity. ^k^ Small mass-gain above 740 °C.

**Table 4 nanomaterials-12-00517-t004:** The ATempH SBR system performance in low-calcium Gamble’s solution and phagolysosomal simulant fluid measured as an average of all dissolution studies presented in this study. The average value ± standard deviation was measured over 24 h.

Test Medium	Gas Flow O_2_ [mL/min]	Gas Flow CO_2_ [mL/min]	Temperature [°C]	pH
Low-calcium Gamble’s solution	144.4 ± 0.9	5.57 ± 0.23	36.7 ± 0.6	7.42 ± 0.14
Phagolysosmal simulant fluid	144.1 ± 0.1	5.59 ± 0.23	36.7 ± 0.3	4.48 ± 0.02

**Table 5 nanomaterials-12-00517-t005:** The mean hydrodynamic size (Z_ave_), polydispersive index (PDI), and zeta potential (ζ_pot_) for nanomaterial dispersions in 0.05% BSA tested in low-calcium Gamble’s solution (*n* = 3) reported as an average of ten repeated measurements ± standard deviation. The three replicates are represented by I, II, and III.

	Z_ave_ [nm]			PDI			ζ_pot_ [mV]		
Nanomaterial	I	II	III	I	II	III	I	II	III
Al_2_O_3_	184.8 ± 1.4	156.6 ± 1.4	165.7 ± 1.1	0.232 ± 0.009	0.162 ± 0.012	0.162 ± 0.013	−21.28 ± 0.59	−21.24 ± 1.28	−22.37 ± 0.65
TiO_2_ (NM-104)	1027.2 ± 228.6	724.0 ± 160.2	1028.7 ± 468.7	0.805 ± 0.145	0.741 ± 0.140	0.719 ± 0.143	−0.880 ± 0.241	−0.840 ± 0.960	0.135 ± 0.302
ZnO (NM-110)	248.7 ± 2.9	247.7 ± 2.7	250.6 ± 1.1	0.146 ± 0.015	0.138 ± 0.016	0.138 ± 0.020	−16.58 ± 0.44	−14.21 ± 0.54	−13.37 ± 0.27
ZnO (NM-111)	279.4 ± 2.7	283.5 ± 2.1	278.9 ± 2.9	0.148 ± 0.015	0.156 ± 0.020	0.155 ± 0.017	−13.78 ± 0.84	−14.72 ± 0.40	−14.46 ± 0.54
ZnO (NM-113)	390.7 ± 5.0	402.1 ± 5.8	244.6 ± 5.8	0.206 ± 0.020	0.203 ± 0.020	0.229 ± 0.009	−6.94 ± 0.49	−6.27 ± 0.49	−7.80 ± 1.01
SiO_2_ (NM-200)	4749.4 ± 773.0	4256.1 ± 991.5	2794 ±4 74.8	0.982 ± 0.053	0.982 ± 0.056	1.00 ± 0.00	−38.21 ± 0.56	−38.45 ± 0.64	−39.20 ± 0.79
CeO_2_ (NM-212)	267.6 ± 4.6	259.6 ± 5.8	244.6 ± 5.8	0.220 ± 0.017	0.216 ± 0.015	0.218 ± 0.011	18.90 ± 0.76	25.81 ± 0.64	29.64 ± 0.93
Bentonite (NM-600)	246.3 ± 8.7	242.7 ± 4.1	242.2 ± 8.3	0.403 ± 0.040	0.368 ± 0.028	0.374 ± 0.030	−43.05 ± 1.51	−42.29 ± 1.42	−44.08 ± 1.41

**Table 6 nanomaterials-12-00517-t006:** Average hydrodynamic size (Z_ave_), zeta potential (ζ_pot_), and polydispersive index (PDI) for nanomaterial dispersions in 0.05% BSA tested in PSF (*n* = 1) reported as an average of ten repeated measurements ± standard deviation.

Nanomaterial	Z_ave_ [nm]	ζ_pot_ [mV]	PDI
Al_2_O_3_	164.7 ± 1.3	−23.52 ± 1.01	0.159 ± 0.016
TiO_2_ (NM-104)	366.7 ± 153.7	−1.33 ± 1.36	0.304 ± 0.095
ZnO (NM-110)	247.1 ± 2.5	−14.55 ± 0.58	0.145 ± 0.019
ZnO (NM-111)	275.9 ± 2.6	−16.73 ± 0.80	0.147 ± 0.023
ZnO (NM-113)	375.8 ± 9.8	−7.55 ± 0.69	0.205 ± 0.018
SiO_2_ (NM-200)	1985.9 ± 886.6	−36.7 ± 0.7	0.945 ± 0.065
CeO_2_ (NM-212)	242.7 ± 4.2	18.52 ± 0.59	0.211 ± 0.017
Bentonite (NM-600)	253.6 ± 5.0	−41.20 ± 1.04	0.352 ± 0.021

**Table 7 nanomaterials-12-00517-t007:** The *p*-values of the reactivity in low-calcium Gamble’s solution. α = 0.05 was used for statistical significance.

Nanomaterial	Parallel	Equal
Al_2_O_3_	0.4149	0.6294
TiO_2_ (NM-104), aluminum coating	0.0597	0.7724
ZnO (NM-110)	0.1227	0.4823
ZnO (NM-111)	<0.0001	N/A
ZnO (NM-113)	0.4184	0.9688
SiO_2_ (NM-200)	0.1080	0.8613
CeO_2_ (NM-212)	0.0021	N/A
Bentonite (NM-600)	0.0034	N/A

**Table 8 nanomaterials-12-00517-t008:** Overview of the calculated initial dissolution rates ${\left(\frac{dC_{A}}{dt}\right)}_{t=0}$ for each repeat (I (■), II (●), and III (▲) of testing and the surface-area dissolution rate, ${\left(\frac{dC\left(SSA\right)}{dt}\right)}_{t=0}$. *p*-values testing the differences between the three repeats using α = 0.05 as the significance level. The dissolution profiles of each material are both tested for parallelism and equality. *p*-value > 0.05 represents no significant difference between the three repeats.

Nanomaterial	Dissolution Rate, ${\left(\frac{dC_{A}}{dt}\right)}_{t=0}$ Repeat I [mg/L/h]	Dissolution Rate, ${\left(\frac{dC_{A}}{dt}\right)}_{t=0}$ Repeat II [mg/L/h]	Dissolution Rate, ${\left(\frac{dC_{A}}{dt}\right)}_{t=0}$ Repeat III [mg/L/h]	Average of the Replicates within All Repeats, Surface Area Dissolution Rate (*n* = 9), ${\left(\frac{dC\left(BET\right)}{dt}\right)}_{t=0}$ [cm^2^/L/s]	Parallel, *p*-Value	Equal, *p*-Value
Al_2_O_3_	0.144 ± 0.080	0.090 ± 0.011	0.115 ± 0.047	0.065 ± 0.029	0.5277	0.4137
TiO_2_ (NM-104), aluminum coating	0.160 ± 0.038	0.159 ± 0.011	0.193 ± 0.022	0.027 ± 0.004	0.0074	ND
ZnO (NM-110)	2.04 ± 0.22	2.24 ± 0.83	5.42 ± 3.36	0.112 ± 0.082	0.6578	<0.0001
ZnO (NM-111)	1.95 ± 0.26	1.50 ± 1.41	1.48 ± 0.61	0.074 ± 0.033	0.0627	0.0051
ZnO (NM-113)	1.73 ± 0.07	2.07 ± 0.50	2.38 ± 0.59	0.036 ± 0.008	0.4210	0.1727
SiO_2_ (NM-200)	3.09 ± 0.10	3.58 ± 0.13	2.92 ± 0.26	3.03 ± 0.317	<0.0001	ND
CeO_2_ (NM-212)	<LOD *	<LOD *	<LOD *	<LOD *	ND	ND
Bentonite (NM-600), release of silicon	0.082 ± 0.028	ND	0.052 ± 0.020	0.096 ± 0.003	0.0052	ND

* CeO_2_ (NM-212) showed no dissolution above LOD within 24 h of dissolution.

**Table 9 nanomaterials-12-00517-t009:** Overview of the calculated initial dissolution rates ${\left(\frac{dC_{A}}{dt}\right)}_{t=0}$ and surface-area dissolution rate ${\left(\frac{dC\left(BET\right)}{dt}\right)}_{t=0}$ of the eight materials in phagolysosomal simulant fluid reported as the average value ± standard deviation.

Nanomaterial	Dissolution Rate, ${\left(\frac{dC_{A}}{dt}\right)}_{t=0}$ [mg/L/h]	${\left(\frac{dC\left(BET\right)}{dt}\right)}_{t=0}$ [cm^2^/L/s]
Al_2_O_3_	0.356 ± 0.001	0.197 ± 0.001
TiO_2_ (NM-104), aluminum coating	0.096 ± 0.002	0.015 ± 2.73 × 10^−4^
ZnO (NM-110)	Highly soluble ^ⱡ^	ND
ZnO (NM-111)	Highly soluble ^ⱡ^	ND
ZnO (NM-113)	Highly soluble ^ⱡ^	ND
SiO_2_ (NM-200)	0.058 ± 3.29 × 10^−3^	0.055 ± 3.12 × 10^−3^
CeO_2_ (NM-212)	0.029 ± 5.03 × 10^−3^	2.20 × 10^−3^ ± 3.80 × 10^−4^
Bentonite (NM-600), release of silicon	0.059 ± 0.013	8.51 × 10^−3^ ± 1.91 × 10^−3^

^ⱡ^ The ZnO materials (NM-110, NM-111, and NM-113) dissolved entirely within the first 25 min. The rate could not be determined as the materials were entirely dissolved at the first sampling time-point. The ZnO materials were therefore described as highly soluble. ^ⱡ^ Highly soluble referred to 100% of the material dissolves within ≤25 min. No quantitative dissolution rates could be determined with the current setup of the ATempH SBR system.

## Data Availability

The data is available from the eNanoMapper database when the embargo of the EU project PATROLS is lifted in October 2023; http://www.enanomapper.net/data.

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
