# Peer review of "Validation and Demonstration of an Atmosphere-Temperature-pH-Controlled Stirred Batch Reactor System for Determination of (Nano)Material Solubility and Dissolution Kinetics in Physiological Simulant Lung Fluids"

_nanomaterials, 2022, doi:10.3390/nano12030517_

Round 1

Reviewer 1 Report

The authors have designed a nice system to evaluate the dissolution behavior of nanoparticles, which  is very creative and practical. The experimental design is very exquisite, and the experimental results are fully explained.The only question is, what is the basis for the chosen of the nanoparticles for test ? Most of the NPs are metal oxide nanoparticles . What about the inorganic nanoparticles  and pure metal nanoparticles . It is recommended to explain it in the introduction.

Author Response

Dear reviewer,

Thanks for your positive assessment of our manuscript. We have tried to address your question below.  

As described in line 123-126, we have chosen well-characterized materials with fast, partial, and slow dissolution and known variation from practically (TiO2) to relatively soluble in different media (silica and ZnO). This has been done to cover relatively wide range in potential dissolution scenarios of manufactured nanomaterials. However, this is an important question, which also was raised by another reviewer. We have combined the two questions and have in the discussion addressed, that the ATempH SBR system can also be used for carbon based (e.g., nanotubes) and pure metal based materials.  

“…The use of the ATempH SBR system is not limited to nanoclays and metal-oxide (nano)materials, the dissolution of, e.g., carbon-based and pure metal-based materials, etc.  can also be conducted. However, we chose well-characterized coated and uncoated OECD materials with slow, partial, and high dissolution rates for validation and demonstration of the ATempH SBR system….”

The manuscript has been proofread by a native American.

Reviewer 2 Report

This manuscript proposes a new method for determination of nanomaterials in simulant lung fluids. There are some issues should be addressed.

  1. Can DLS detect nanomaterial solubility and dissolution kinetics?
  2. What are the advantages of the proposed system compared with the reported methods? Please provide the data of comparison.
  3. Can the system detect other type nanomaterials except for metal-based nanomaterials? Please clarify this issue.
  4. The quantitive data are lacking in the abstract.

Author Response

Dear reviewer,

Thanks for your positive assessment of our manuscript. We have tried to address your comments below:

1. Can DLS detect nanomaterial solubility and dissolution kinetics?

DLS was in this study only used to evaluate and document the quality of nanomaterial dispersions. We used ICP-MS to determine the total amount of dissolved ions. DLS can not be used to determine dissolution rate for these materials. DLS could be used by proxy for fast dissolving and highly stable suspensions with non-agglomerated near-spherical particles. We have previously tried to add DLS with flow-cuvette for simultaneous in-line monitoring of hydrodynamic data and dissolution, but failed due to problems with accumulation and agglomeration of material in the cuvette and too little sensitivity to this type of material.

2. What are the advantages of the proposed system compared with the reported methods? Please provide the data of comparison.

The advantage of the ATempH SBR system is the precise control of Atmosphere, Temperature and pH as well as online measurement of redox potential, which other dissolution systems for testing nanomaterials do not provide, as presented in the introduction (line 117-121). It also allows for assessment of initial dissolution behavior, which is also important for assessment of potential acute clearance and effects.

Limited dissolution data has previously been reported in phagolysosomal simulant fluid, and to the best of our knowledge, no publications consider dissolution of the selected nanomaterials in low-calcium Gamble’s solution. A direct comparison between existing methods were above the scope of this study, as we only consider validation and demonstration of a new static dissolution system. However, a comparison between static and dynamic systems of the same (nano)materials will be conducted in future studies. Existing data using the continuous flow system report on long-term dissolution rates while we analyzed initial and short-term rates for the method validation. Longer durations can also be studied.

3. Can the system detect other type nanomaterials except for metal-based nanomaterials? Please clarify this issue.

This question was also raised by another reviewer. We therefore combine the two questions related to dissolution testing of non-metal-oxide nanomaterials. The system can be used to study dissolution of other types of nanomaterials, however, these were not included in the validation of the system. The type of material that can be tested relies in principle on whether it can be analyzed chemically or not. For the validation, we have chosen different well-characterized types of nanomaterials (metal-oxides and a nanoclay) with fast, partial, and slow dissolution to show that the potential dissolution scenarios. The choice of materials were described in line 123-126. However, the question regarding the applicability to other than metal-oxide materials is relevant to address, which we have done in the discussion.

“…The use of the ATempH SBR system is not limited to nanoclays and metal-oxide (nano)materials, the dissolution of, e.g., carbon-based and pure metal-based materials, etc.  can also be conducted. However, we chose well-characterized coated and uncoated OECD materials with slow, partial, and high dissolution rates for validation and demonstration of the ATempH SBR system….”

4. The quantitive data are lacking in the abstract.

Thanks for addressing this question. We have rewritten parts of the abstract, which now includes p-values for the materials demonstrating statistical identical dissolution across the repeats (I, II, and III).

“..and bentonite (NM-600) showing high intralaboratory repeatability and robustness across repeated testing (I, II, and III) in triplicate (replicate 1, 2, and 3) in low-calcium Gamble’s solution. A two-way repeated-measures ANOVA was used to determine the intralaboratory repeatability in low-calcium Gamble’s solution, where Al2O3 (p=0.5277), ZnO (NM-110, p=0.6578), ZnO (NM-111, p=0.0627), ZnO (NM-113, p=0.4210) showed statistical identical repeatability across repeated testing (I, II, and III). The dissolution of the materials were also tested in PSF to demonstrate the applicability of the ATempH SBR system in other physiological fluids. We further show the uncertainty levels at which dissolution can be determined using the ATempH SBR system…”